# Active Noise Control over Space: A Subspace Method for Performance Analysis

**Jihui Zhang** [1,*], **Thushara D. Abhayapala** [1], **Wen Zhang** [1,2] **and Prasanga N. Samarasinghe** [1]

1   Audio & Acoustic Signal Processing Group, College of Engineering and Computer Science, Australian National University, Canberra 2601, Australia; thushara.abhayapala@anu.edu.au (T.D.A.); wen.zhang@nwpu.edu.cn (W.Z.); prasanga.samarasinghe@anu.edu.au (P.N.S.)
2   Center of Intelligent Acoustics and Immersive Communications, School of Marine Science and Technology, Northwestern Polytechnical University, Xi'an 710072, China
*   Correspondence: jihui.zhang@anu.edu.au

**Abstract:** In this paper, we investigate the maximum active noise control performance over a three-dimensional (3-D) spatial space, for a given set of secondary sources in a particular environment. We first formulate the spatial active noise control (ANC) problem in a 3-D room. Then we discuss a wave-domain least squares method by matching the secondary noise field to the primary noise field in the wave domain. Furthermore, we extract the subspace from wave-domain coefficients of the secondary paths and propose a subspace method by matching the secondary noise field to the projection of primary noise field in the subspace. Simulation results demonstrate the effectiveness of the proposed algorithms by comparison between the wave-domain least squares method and the subspace method, more specifically the energy of the loudspeaker driving signals, noise reduction inside the region, and residual noise field outside the region. We also investigate the ANC performance under different loudspeaker configurations and noise source positions.

**Keywords:** active noise control (ANC); performance analysis; wave domain; spatial noise; reverberant room

## 1. Introduction

Active noise control (ANC) over a spatially extended region, which is termed as *'Spatial ANC'*, is a challenging field of research as the aim is to create a large quiet zone for multiple listeners in three-dimensional (3-D) spaces. In spatial ANC applications, such as noise cancellation in aircraft [1] and automobiles [2–5], multichannel ANC systems equipped with multiple sensors and multiple secondary sources are adopted [6]. In literature, both time-domain [7,8] and frequency-domain [9,10] algorithms have been implemented in multichannel ANC systems, which can cancel the noise at error sensor positions and their close surroundings [10]. Recently, ANC over space has been approached via Wave field synthesis (WFS)-based wave-domain algorithms [11–13] and (cylindrical/spherical) harmonic-based wave-domain algorithms [14–19], with which the noise over entire region of interest can be cancelled directly. Here onwards, we use the terminology *'wave-domain ANC'* to refer to harmonics-based wave-domain ANC.

In wave-domain ANC, the number of secondary sources/loudspeakers is required to be no less than the number of modes in the spatial region, so that all the modes can be controlled. When the number of loudspeakers cannot control all the modes in the spatial region, using the wave-domain adaptive algorithms and conventional adaptive algorithms, the noise reduction performance in the steady state degrades significantly [16]. In practical applications, numbers and locations of the

loudspeakers have more constraints compared to simulation setups in [16]. For instance, in a vehicle, the numbers and positions of the loudspeakers are highly constrained by the vehicle dimensions and passenger convenience. It is valuable for design engineers to estimate whether the available numbers and positions of the loudspeakers are sufficient to the noise reduction requirements, before they implement an ANC system in a real environment.

Since the noise reduction performance varies with different ANC algorithms, it is important to investigate the maximum achievable performance for the given system, which is dependent on the coherence between the reference sensors and the error sensors [20], secondary source characteristics and locations, room environments, and the primary noise field characteristics.

In the literature, Laugesen et al. provided the theoretical prediction for multichannel ANC in a small reverberant room based on the primary signal on the error sensors [21]. For spatial ANC performance estimation over an entire region, Chen et al. investigated ANC performance by noise pattern analysis of the primary noise field [22,23]. Buerger et al. investigated the coherence between two observation points in the noise field evoked by given continuous source distributions, which can be applied to predict the upper bound of ANC performance in the region of interest [24]. However, the capability of secondary sources, in particular room environments, has not yet been explored.

In this paper, we investigate the maximum noise control performance over a spatial region by investigating capability of secondary sources, in particular room environments. We define the subspace spanned by the wave-domain secondary-path coefficients, and evaluate the ANC performance in the subspace. Using the proposed subspace method, the design engineers can predict the noise cancellation performance before an actual ANC system is implemented in a product. Simulations are conducted in a 3-D room environment under different noise source positions, when the loudspeakers have constraints on numbers and positions. The proposed subspace method is more feasible than the wave-domain least squares method.

The rest of this paper is organized as follows. In Section 2, we formulate the ANC problem in a 3-D room. We investigate the maximum ANC performance using the wave-domain least squares method in Section 3, and investigate the maximum ANC performance using the subspace method in Section 4. The simulation validation is demonstrated in Section 5. We draw some conclusions in Section 6.

## 2. Problem Formulation

In this section, we formulate the ANC problem in the wave domain to cancel the noise over a 3-D spatial region inside rooms. As shown in Figure 1, let the quiet zone of interest (blue area) be a spherical region (S) with a radius $R_1$. Assume that the noise sources (black stars) and the secondary sources (black loudspeakers) are located outside the region of interest. In the ANC process, we measure the noise field by placing a spherical microphone array (dark blue stars) on the boundary of the control region (i.e., the desired quiet zone).

Any arbitrary observation point within the control region is denoted as $x \equiv \{r, \phi_x, \psi_x\}$. Here, $\psi$ and $\phi$ are the elevation angle and the azimuthal angle, respectively. In the ANC system, the residual signal at this point $e(x, k)$ is given by

$$e(x, k) = v(x, k) + s(x, k), \tag{1}$$

where $k = 2\pi f / c$ is the wave number, $f$ is the frequency, and $c$ is the speed of sound propagation. $v(x, k)$ and $s(x, k)$ denote the primary noise field and the secondary noise field observed at point $x$, respectively.

The secondary noise field generated by a discrete loudspeaker array with $L$ loudspeakers (A simple model of the loudspeaker is applied here. More models characterizing the acoustic, mechanic and electric system of the loudspeakers are investigated in [25].) can be represented by

$$s(x, k) = \sum_{l=1}^{L} d_l(k) G(x|y_l, k),$$
(2)

where $d_l(k)$ is the driving signal for the $l^{\text{th}}$ loudspeaker, $y_l$ denotes the location of the $l^{\text{th}}$ loudspeaker, and $G(x|y_l, k)$ denotes the acoustic transfer function (ATF) between the $l^{\text{th}}$ loudspeaker and the observation point $x$. Please note that in the reverberant environment, $G(x|y_l, k)$ includes the room reflections.

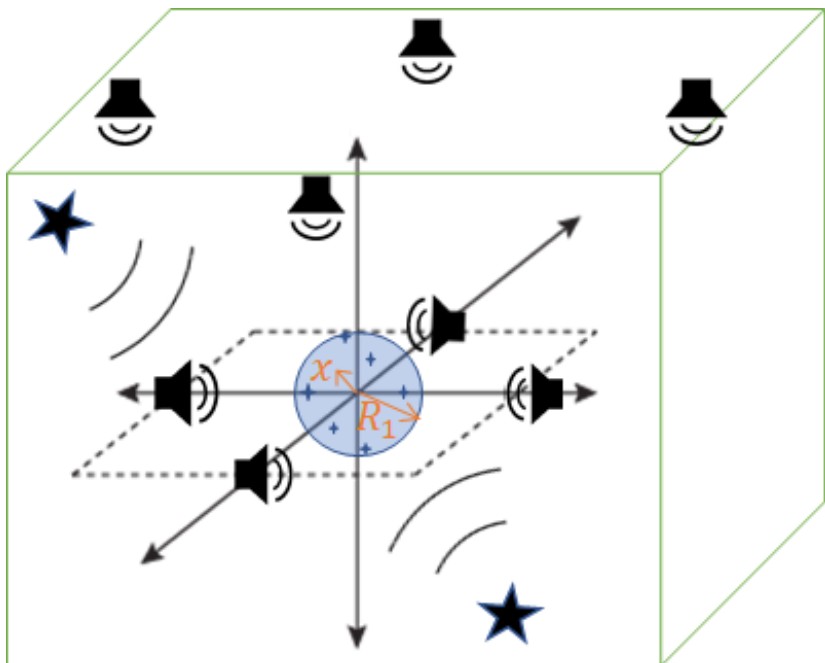

**Figure 1.** ANC system in a 3-D room. Black stars represent primary sources, loudspeakers represent secondary sources, the blue sphere represents the control region, and dark blue stars represent error microphones over the control region.

Instead of using measurements of the microphone signals directly, the wave-domain approach employs the wave equation solutions as basis functions to express any wave field over the spatial region, and designs the secondary noise field based on the wave-domain decomposition coefficients. We then represent the primary noise field and secondary noise field in the wave domain.

The spherical harmonics-based wave equation solution decomposes any homogeneous incident wave field $v(x, k)$ observed at $x$ into

$$v(x, k) = \sum_{u=0}^{\infty} \sum_{m=-u}^{u} \beta_{um}(k) j_u(kr) Y_{um}(\phi_x, \psi_x),$$
(3)

where $j_u(\cdot)$ is the spherical Bessel function of order $u$ and $Y_{um}(\cdot)$ denotes the spherical harmonics. Therefore, the decomposition coefficients $\beta_{um}(k)$ represent the primary noise field in the wave domain.

Within the region of interest $r \leq R_1$, a finite number of modes can be used to approximate the noise field [26]. Thus, the primary noise field in (3) can be truncated by

$$v(x, k) \approx \sum_{u=0}^{N} \sum_{m=-u}^{u} \beta_{um}(k) j_u(kr) Y_{um}(\phi_x, \psi_x),$$
(4)

where the truncation order of $N = \lceil ekR_1/2 \rceil$ [26–28]. In the vector version, $\boldsymbol{\beta}(k) = [\beta_{0,0}(k), \beta_{1,-1}(k), \ldots, \beta_{N,N}(k)]^T$.

Using the spherical harmonic expansion, the secondary noise field within the quiet zone can also be represented by

$$s(\boldsymbol{x}, k) = \sum_{u=0}^{\infty} \sum_{m=-u}^{u} \gamma_{um}(k) j_u(kr) Y_{um}(\phi_{\boldsymbol{x}}, \psi_{\boldsymbol{x}}), \tag{5}$$

where the coefficients $\gamma_{um}(k)$ represent the secondary noise field in the wave domain.

Similar to the primary noise field, inside the region of interest with the radius of $R_1$, the secondary noise field can be truncated by

$$s(\boldsymbol{x}, k) \approx \sum_{u=0}^{N} \sum_{m=-u}^{u} \gamma_{um}(k) j_u(kr) Y_{um}(\phi_{\boldsymbol{x}}, \psi_{\boldsymbol{x}}). \tag{6}$$

The ATF in (2) can be parameterized in the wave domain [29] as

$$G(\boldsymbol{x}|\boldsymbol{y}_l, k) \approx \sum_{u=0}^{N} \sum_{m=-u}^{u} \eta_{um}^{(l)}(k) j_u(kr) Y_{um}(\phi_{\boldsymbol{x}}, \psi_{\boldsymbol{x}}), \tag{7}$$

where $\eta_{um}^{(l)}(k)$ is the ATF in wave domain for each loudspeaker.

Substituting (6) and (7) into (2), the secondary sound coefficients $\gamma_{um}(k)$ can also be represented by

$$\gamma_{um}(k) = \sum_{l=1}^{L} d_l(k) \eta_{um}^{(l)}(k). \tag{8}$$

In matrix form, the relationship between the secondary source decomposition coefficients and the loudspeaker driving signals is given by

$$\boldsymbol{\gamma}(k) = \boldsymbol{\eta}(k) \boldsymbol{d}(k), \tag{9}$$

where

$$\boldsymbol{\eta}(k) = \begin{bmatrix} \eta_{00}^{(1)}(k) & \eta_{00}^{(2)}(k) & \cdots & \eta_{00}^{(L)}(k) \\ \eta_{-11}^{(1)}(k) & \eta_{-11}^{(2)}(k) & \cdots & \eta_{-11}^{(L)}(k) \\ \vdots & \vdots & \ddots & \vdots \\ \eta_{NN}^{(1)}(k) & \eta_{NN}^{(2)}(k) & \cdots & \eta_{NN}^{(L)}(k) \end{bmatrix}, \tag{10}$$

and $\boldsymbol{d}(k) = [d_1(k), \ldots, d_L(k)]^T$.

We investigate the maximum achievable ANC performance based on the primary noise field coefficients $\boldsymbol{\beta}(k)$ and the secondary-path information $\boldsymbol{\eta}(k)$ in the wave domain, and derive loudspeaker driving signals using two methods: the wave-domain least squares method and the subspace method.

## 3. Wave-Domain Least Squares Method

One method for deriving the loudspeaker driving signal $\boldsymbol{d}(k)$ is to match the secondary noise field coefficients to the primary noise field coefficients, in the region of interest. Therefore,

$$\boldsymbol{\eta}(k) \boldsymbol{d}(k) = -\boldsymbol{\beta}(k). \tag{11}$$

Here, we denote the set of all linear combinations of the columns in $\boldsymbol{\eta}(k)$ as column space $\boldsymbol{C}$. The rank of matrix $\boldsymbol{\eta}(k)$ defines the dimension of the column space $\boldsymbol{C}$.

In Equation (11), (i) the number of loudspeakers, (ii) the rank of $\boldsymbol{\eta}(k)$ and (iii) the rank of matrix $\boldsymbol{\eta}(k)$ augmented by $\boldsymbol{\beta}(k)$ specify the number of solutions for the linear system (11).

- Case 1: $L = (N+1)^2$

If the number of loudspeakers is same as the number of modes in the region of interest, (11) has one unique solution, which is given by

$$d(k) = -(\eta(k))^{-1}\beta(k), \tag{12}$$

where $(\cdot)^{-1}$ denotes the inverse of a matrix.

- Case 2: $L > (N+1)^2$

If the number of loudspeakers is greater than the mode requirement, (11) is an under-determined system. There is either no solution or an infinite number of solutions. In practice, however, this case rarely happens, as extra loudspeakers do not result in better ANC results but increase the device cost and computational cost.

- Case 3: $L < (N+1)^2$

If the loudspeaker number $L$ is less than the number of modes $(N+1)^2$ in the region of interest, (11) is an over-determined system. There are more equations than unknowns, resulting in either a single unique solution or no solution.

When the measurements are in a very special case, which requires

$$\text{rank}(\eta|\beta) = \text{rank}(\eta), \tag{13}$$

Equation (11) has an exact solution. Here, $\text{rank}(\eta)$ denotes the rank of $\eta$, and $\text{rank}(\eta|\beta)$ denotes the rank of the matrix $\eta(k)$ augmented by $\beta(k)$.

When

$$\text{rank}(\eta|\beta) \neq \text{rank}(\eta), \tag{14}$$

Equation (11) has no exact solution. The solution can be approximated using the least squares method [30]. The least squares method tries to find the best approximation which results in minimum mean square errors [31], by solving the following problem

$$\min\|\eta(k)d(k) - (-\beta(k))\|^2. \tag{15}$$

The optimal solution of this minimization problem can be written as

$$d(k) = -(\eta(k))^{\dagger}\beta(k), \tag{16}$$

where $(\cdot)^{\dagger}$ denotes the pseudoinverse of a matrix.

For some circumstances, $\beta(k)$ could be totally inside the column space $C$. While most of time, $\beta(k)$ have components outside the space $C$. In general, the result of the driving signal in (16) is achieved by solving the equation as follows:

$$\eta(k)d(k) = -\text{Proj}_C\beta(k), \tag{17}$$

where $\text{Proj}_C\beta(k)$ denotes the projected part of the primary noise field coefficients $\beta(k)$ in the column space $C$. The projection matrix can be also written by

$$\text{Proj}_C\beta(k) = \eta(k)(\eta^H(k)\eta(k))^{-1}\eta^H(k)\beta(k). \tag{18}$$

In most applications, the number of loudspeakers is less than the requirement. Then the driving signals can be designed by least squares solutions in (16). Therefore, this method is here called the 'wave-domain least squares method' (WDLS).

## 4. Subspace Method

In the second method, we obtain the secondary-path coefficient $\eta^{(l)}(k)$, extract the subspace spanned by secondary-path coefficients which represents the secondary sources in this environment, and only cancel the primary noise field which can be projected into this subspace.

### 4.1. Principal Component Analysis of the Secondary Path

Let $\eta^{(l)}(k) = [\eta_{00}^{(l)}(k), \eta_{-11}^{(l)}(k), \ldots, \eta_{NN}^{(l)}(k)]^T$ the wave-domain secondary-path coefficients for the $l_{\text{th}}$ loudspeaker. Matrix $\eta(k)$ in (10) for the entire loudspeaker array represents the secondary path, where

$$\eta(k) = [\eta^{(1)}(k), \ldots, \eta^{(L)}(k)]. \tag{19}$$

In an arbitrary loudspeaker array setup, each column of matrix $\eta(k)$ is not necessarily orthogonal. We perform the principal component analysis (PCA) of the correlation matrix $E\{\eta^H(k)\eta(k)\}$ to obtain an orthonormal eigen-basis for the space of the secondary path in the wave domain.

We take the correlation matrix $E\{\eta^H(k)\eta(k)\}$, and then decompose this matrix into a set of orthonormal eigenvectors and their corresponding eigenvalues, as follows:

$$E\{\eta^H\eta\} = u\lambda v, \tag{20}$$

where $u = [u_1, \ldots, u_i, \ldots, u_L]$ are the eigenvectors of the wave-domain ATF, $v = u^T$, and the $i^{\text{th}}$ column corresponds to the eigenvalue $\lambda_i$. Here onwards, the frequency dependent $k$ is omitted for notational simplicity. The eigenvalues in the matrix form are

$$\lambda = \begin{bmatrix} \lambda_1 & 0 & \cdots & 0 \\ 0 & \lambda_2 & \cdots & 0 \\ \vdots & \vdots & \ddots & \vdots \\ 0 & 0 & \cdots & \lambda_L \end{bmatrix}. \tag{21}$$

Here, the vectors $u_1, \ldots, u_i, \ldots, u_L$ are written in order of descending eigenvalues $\lambda$ [32]. On this basis $u$, the first few largest eigenvalues correspond to the principal components of the secondary path in wave domain ($\eta$), which contain the most useful information.

Depending on the acoustic environment and the loudspeaker placement, the first $B$ components are used to represent the loudspeakers, then the corresponding eigenvectors are $u^\diamond = [u_1, \ldots, u_B]$. The subspace $O$ spanned by the wave-domain secondary-path coefficients $\eta$ is defined as

$$O = \eta u^\diamond, \tag{22}$$

where the dimension of $u^\diamond$ is $L \times B$, and $B \leq L$.

By normalizing each column of matrix $O$, the orthonormal vectors $o_1, \ldots, o_B$ are obtained. These vectors generate a subspace, which represents the loudspeaker array and the acoustic environment. The dimensions of basis $O$ are $(N+1)^2 \times B$.

For the $l^{\text{th}}$ loudspeaker, the average ATF coefficients over certain short frames can be represented in this space as

$$\bar{\eta}^{(l)} = \sum_{b=1}^{B} \kappa_b^{(l)} o_b, \tag{23}$$

where $\kappa_b^{(l)}$ are the projection coefficients. In vector form, (23) can be written by

$$\bar{\eta}^{(l)} = O\kappa^{(l)}, \tag{24}$$

where $\kappa^{(l)} = \{\kappa_1^{(l)}, \ldots, \kappa_b^{(l)}, \ldots, \kappa_B^{(l)}\}^T$ and

$$\kappa_b^{(l)} = < \eta^{(l)}, o_b > . \tag{25}$$

Here, $< \cdot, \cdot >$ is the inner product of two vectors.

Therefore, $\kappa = \{\kappa^{(1)}, \ldots, \kappa^{(L)}\}$ is the secondary-path coefficients in the subspace $O$, with the dimension of $B \times L$.

### 4.2. Projection of the Primary Noise Field into the Subspace

Below we project the wave-domain coefficients of the primary noise field into the subspace $O$.

For a new primary noise field represented by vector $\beta$, by projecting $\beta$ into the subspace $O$, we can obtain

$$\text{Proj}_O \beta = \sum_{b=1}^{B} < \beta, o_b > o_b = < \beta, o_1 > o_1 + \cdots + < \beta, o_B > o_B, \tag{26}$$

where $\text{Proj}_O \beta$ denotes the projection of vector $\beta$ into subspace $O$. The matrix form of the projection is represented by

$$\text{Proj}_O \beta = O y, \tag{27}$$

where $y = \{y_1, y_2, \ldots, y_B\}^T$ are the primary noise field coefficients in the subspace, and $y_b = < \beta, o_b >$.

Therefore, the primary noise field can be separated by two parts: the projected part and the remaining part,

$$\beta = \text{Proj}_O \beta + R(\beta), \tag{28}$$

where $R(\beta)$ is the orthogonal complement of the subspace $O$. The projected part indicates the primary noise field which can be cancelled in this system setup, and the orthogonal complement indicates the primary noise field which cannot be cancelled in this system.

If $R(\beta) = 0$, $\beta$ lies in the subspace, then the primary noise field can be completely cancelled by the loudspeaker array.

In more general cases, $R(\beta) \neq 0$. This indicates the limitation of noise cancellation over the region of interest, under the particular loudspeaker placement and acoustic environment.

Next, we design the driving signal of loudspeaker $d_l(k)$ to cancel the primary noise field projected into the subspace ($\text{Proj}_O \beta$).

### 4.3. Noise Control in the Subspace

In the subspace, matching the secondary sound field coefficients to the projected primary noise field coefficients, the optimal solution of the secondary noise field coefficients can be written by

$$\gamma = -\text{Proj}_O \beta. \tag{29}$$

The projection from the primary noise field into the loudspeaker subspace $\text{Proj}_O \beta$ can be calculated by (27).

In a given loudspeaker setup, substituting (24) into (9), the representation of secondary noise field coefficients can be rewritten by

$$\gamma = O \kappa d, \tag{30}$$

where $d = \{d_1, \ldots, d_L\}^T$, $\kappa = \{\kappa^{(1)}, \ldots, \kappa^{(L)}\}$.

Substituting (27) and (30) into (29), the final equation to design the driving signal can be written as

$$\kappa d = -y. \tag{31}$$

The loudspeaker driving signals can be calculated by solving the system of linear equations described by (31). The number of principal components specifies whether the linear system (31) can be solved exactly.

- Case 1: $B = L$

  When we reserve all the information in the PCA, (31) has only one unique solution. In that case, the driving signals can be represented by

  $$d = -(\kappa)^{-1}y. \tag{32}$$

- Case 2: $B < L$

  When we only use the largest components to generate the subspaces, instead of solving the over-determined system in (11), Equation (31) solves an under-determined system. In that case, the driving signals $d$ can be derived by

  $$d = -(\kappa)^{\dagger}y, \tag{33}$$

  where $(\kappa)^{\dagger}$ is the pseudoinverse of the secondary-path coefficients in the subspace, with the dimension of $L \times B$.

In all cases, loudspeaker driving signals $d$ are designed by the secondary-path information in the subspace $\kappa$ and the primary noise field coefficients in the subspace $y$, as shown in (32) and (33). Therefore, the method is here called the 'subspace method'.

## 5. Simulation Results

In this section, we conduct simulations to investigate the maximum achievable ANC performance in the 3-D sound field by using the WDLS and the proposed subspace method.

When the driving signal is unit amplitude, and only the $l^{\text{th}}$ loudspeaker produces sound, $\eta_{um}^{(l)}(k) = \gamma_{um}^{(l)}(k)$. Therefore, we can capture $\eta_{um}^{(l)}(k)$ from the measurement of $s(x,k)$ based on (6). For the WDLS method, the driving signals can be designed by the solutions of (11). For the subspace method, following the PCA, from (22), we can extract the subspace $O$ from the loudspeaker coefficients $\eta_{um}^{(l)}(k)$. Representing the $\eta_{um}^{(l)}(k)$ in the subspace as $\kappa$, and projecting the primary source into the subspace as $y$, we can derive the driving signals by solving (31).

### 5.1. Simulation Setup

In this simulation, we investigate the ANC performance in the reverberant environment. The reverberant environment is modelled as a cuboid room of 6 m $\times$ 6 m $\times$ 5 m. The reflection coefficients are set to [0.75, 0.8, 0.77, 0.85, 0.1, 0.1]. The reverberation is simulated by the image source method with the image order of 5. The origin of the room is on the left bottom corner. The region of interest is a spherical area with a radius of 0.5 m, and the center of the region is (3, 3, 1.5) with respect to the origin.

We assume that the noise field only contains a single-frequency component. In the following investigations, the primary noise field is a spherical wave coming from a point source located at $(r, \phi, \psi)$ with respect to the center of the region, with a constant magnitude of 10. The locations of primary sources are varying in each case, as shown in Table 1.

We assume the frequency of the noise field is 200 Hz. From (4), the region of interest in such a noise field can be represented by $N = \lceil ekR_1/2 \rceil = 3$ modes. Thus, at least $(N+1)^2 = 16$ microphones must be placed on the boundary to capture the information of the residual noise field for each mode. In this simulation, we place 32 microphones on the spherical boundary, following the Gauss-Legendre sampling method. White Gaussian noise is added to each microphone recording to simulate the internal thermal noise of microphones.

**Table 1.** Loudspeaker array setup and noise source location in 4 cases, which are given in Figures 2 and 5, respectively.

| Noise Source Position / Loudspeaker Array | Non-Symmetry | Symmetry |
|---|---|---|
| $(2, 315°, 45°)$ | case 1 | |
| $(2, 315°, 90°)$ | case 2 | |
| 24 position candidates | case 3 | case 4 |

To control all the modes, 16 loudspeakers are required to be placed outside the control region. To emulate a practical scenario, however, in this simulation, only 12 loudspeakers are used. Among them, 8 loudspeakers are placed in the x-y plane with two different geometries, as shown in Table 1. Another four loudspeakers are placed on another plane, which is on the plane close to the ceiling. The loudspeaker positions for each case are shown in Table 2.

**Table 2.** Loudspeaker positions for non-symmetric placement and symmetric placement.

| | Loudspeakers in the x-y Plane | | | Loudspeakers Outside the x-y Plane | |
|---|---|---|---|---|---|
| No. | Non-Symmetry | Symmetry | No. | Non-Symmetry or Symmetry | |
| 1 | (4, 3, 2.5) | (4.5, 3, 2.5) | 9 | (0.5, 0.5, 4.5) | |
| 2 | (1.8, 3, 2.5) | (1.5, 3, 2.5) | 10 | (5.5, 5.5, 4.5) | |
| 3 | (3, 2, 2.5) | (3, 1.5, 2.5) | 11 | (5.5, 0.5, 4.5) | |
| 4 | (3, 4.2, 2.5) | (3, 4.5, 2.5) | 12 | (0.5, 5.5, 4.5) | |
| 5 | (4.3, 3.2, 2.5) | (4.2, 1.8, 2.5) | | | |
| 6 | (1.7, 2.8, 2.5) | (1.8, 1.8, 2.5) | | | |
| 7 | (3.2, 1.7, 2.5) | (4.2, 4.2, 2.5) | | | |
| 8 | (2.8, 4.2, 2.5) | (1.8, 4.2, 2.5) | | | |

We evaluate the ANC performance in terms of residual noise field, noise reduction on the sample points inside the region $N_r^{in}$, and the energy of the driving signals $E_d$.

To evaluate the actual noise reduction performance within the control region, residual sound fields $e_{in}$ at $Z = 1296$ points uniformly placed within the cross section between the region of interest and the x-y plane are examined. The noise reduction inside the region of interest over $Z$ points $N_r^{in}$ can be written by

$$N_r^{in} \triangleq 10 \log_{10} \frac{\sum_z E\{|e_{in\_z}|^2\}}{\sum_z E\{|e_{in\_z}(0)|^2\}}, \tag{34}$$

where $E\{|e_{in\_z}(0)|^2\}$ is the energy of the primary noise field at the $z^{th}$ sample point, and $E\{|e_{in\_z}|^2\}$ is the energy of the residual noise field at the $z^{th}$ sample point.

To evaluate the loudspeaker energy consumption, we compare the total energy of all the loudspeakers $E_d$. The loudspeaker energy consumption can be represented by

$$E_d = d^T d. \tag{35}$$

For the subspace method, during the PCA process for the secondary path, we only reserve the principle components in $u$. Components in $u$ which correspond to $\lambda_i$ occupying less than 5% of the largest eigenvalue $\lambda_1$ are omitted in $u^\diamond$.

*5.2. Cancellation Performance Using Different Methods*

We first compare cancellation performance using the subspace method and the WDLS method for two different noise source positions. In this simulation, white Gaussian noise with signal to noise ratio (SNR) (Here, the SNR level is with respect to the primary noise field level on virtual microphone in the center of the region.) of 60 dB is added to each microphone recording. For simplicity of plotting,

the cancellation performance over the region is confined to horizontal planes at elevation of 90°
(x-y plane) of the 3-D region.

As shown in Figure 2, we assume noise field is generated by a point source located at (2, 315°,
45°) (case 1) or (2, 315°, 90°) (case 2) in the spherical coordinates. The geometry of loudspeakers in the
x-y plane is not symmetrical. The energy of the primary noise field is shown in Figure 3.

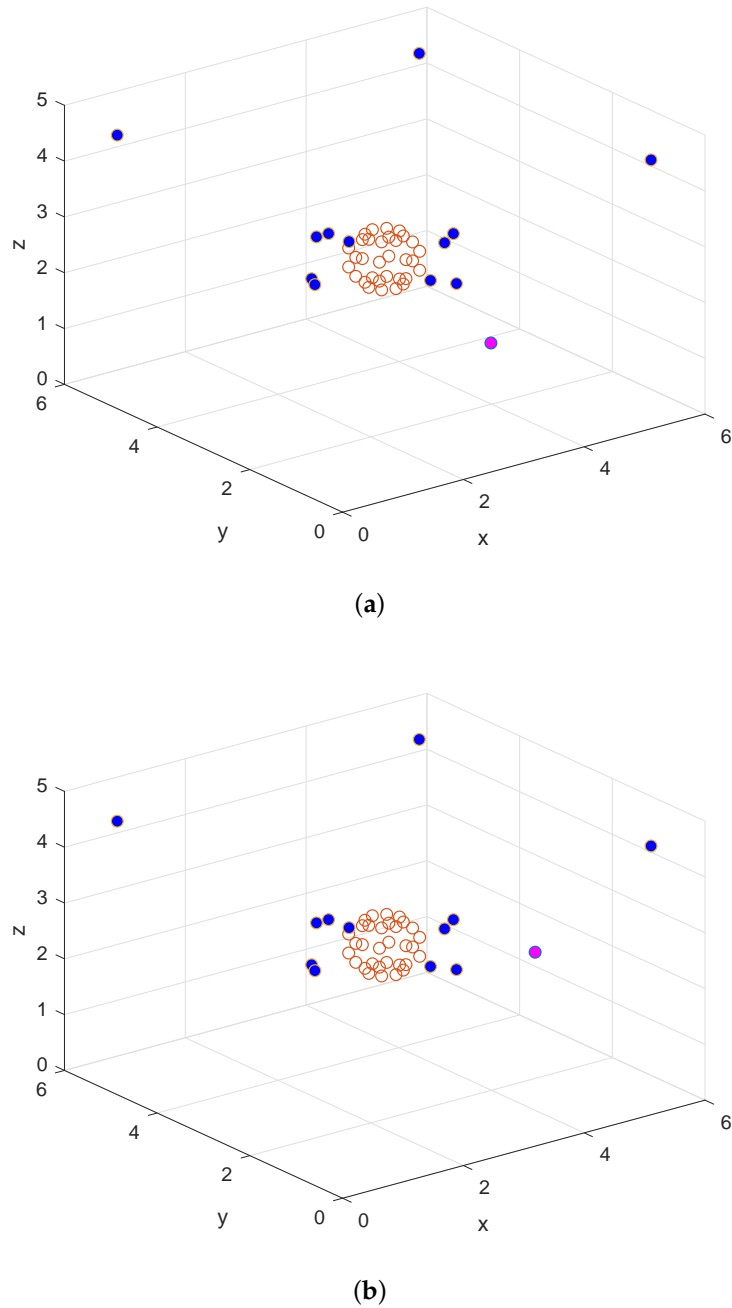

**(a)**

**(b)**

**Figure 2.** ANC system setup, where the pink point is the noise source position, blue points are
loudspeaker positions, and red points are microphone positions: (**a**) case 1; (**b**) case 2.

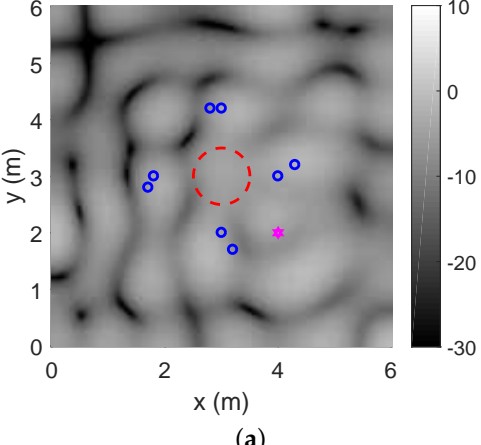
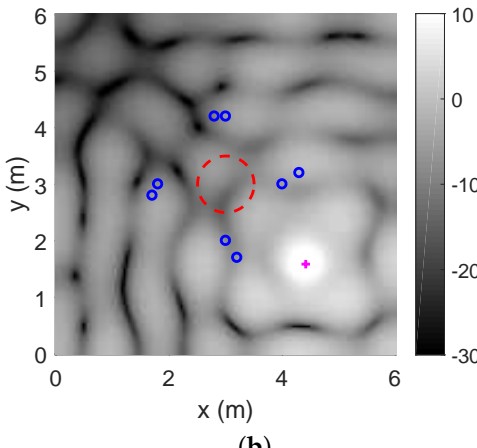

(**a**)　　　　　　　　　　　　　　　　　　　　　　　　　　(**b**)

**Figure 3.** Energy of the primary noise field, where pink point is the projection of the primary source on the x-y plane, blue points are the loudspeaker points located on the x-y plane, and the red dashed circle is the boundary of the region of interest: (**a**) case 1; (**b**) case 2.

Figure 4 demonstrates the energy of the residual noise field in the x-y plane. As we expected, since the number of loudspeakers (12) cannot cover all the modes (16) in the region, in all four figures, the primary noise field in the region of interest cannot be fully cancelled. In case 2, compared with the primary noise field (Figure 3b), both the WDLS method and the subspace method can achieve significant noise reduction in the region of interest, which are dark areas in the middle of Figure 4b,d. In case 1, since the noise source is in a different hemisphere from the loudspeaker array, compared with Figure 3a, cancellation performance inside the region is fairly limited for both WDLS and the subspace method, as shown in Figure 4a,c.

Meanwhile, compared with Figure 4a,b, in Figure 4c,d, the subspace method results in lower energy of the residual noise field outside the region of interest. The WDLS method on the other hand produces increased sound energy outside the region, especially when the noise reduction level is fairly limited inside the region. Using the subspace method, we analyze physically achievable performance and avoid sound amplification outside the control area.

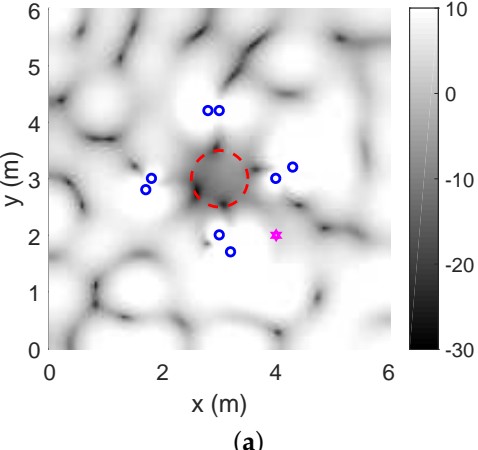
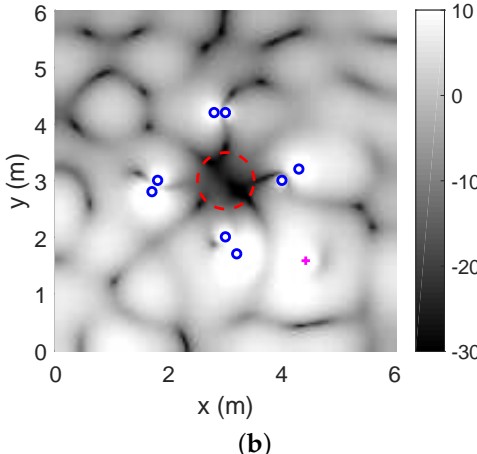

(**a**)　　　　　　　　　　　　　　　　　　　　　　　　　　(**b**)

**Figure 4.** *Cont.*

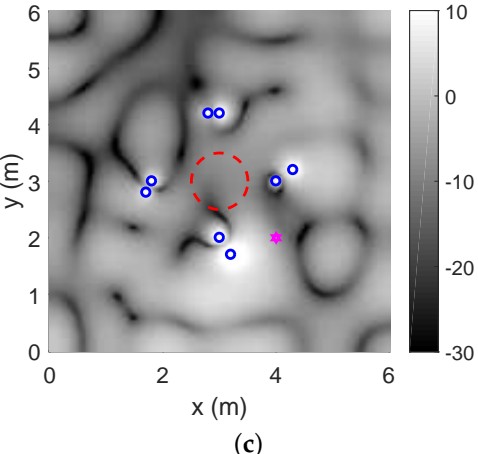

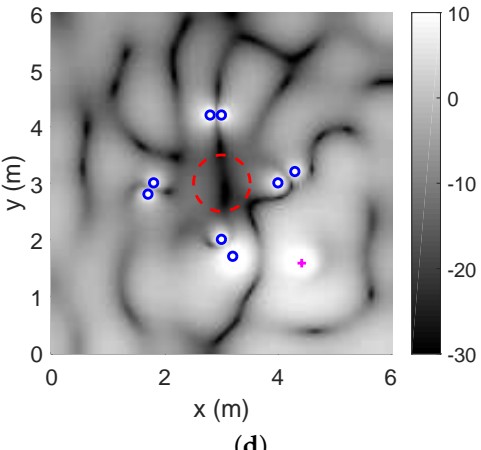

**Figure 4.** Energy of the residual noise field, when the noise field is generated by one primary source using different methods. Here, pink point is the projection of the primary source on the x-y plane and blue points are the loudspeaker points located on the x-y plane: (**a**) the WDLS method in case 1; (**b**) the WDLS method in case 2; (**c**) the subspace method in case 1; (**d**) the subspace method in case 2.

*5.3. Comparison of the Effect of Different Noise Source Positions*

After investigating the cancellation performance in two different noise source positions, we move the noise source position to different elevations (45°, 90°, 135°) and different azimuthal angles (0°, 45°, 90°, 135°, 180°, 225°, 270°, 315°). As shown in Figure 5a, 24 source position candidates are chosen on the sphere with radius of 2 m. We measure the primary noise field coefficients and secondary noise field coefficients by microphones with different SNR levels, which are 60 dB and 30 dB, respectively, for each source position candidate.

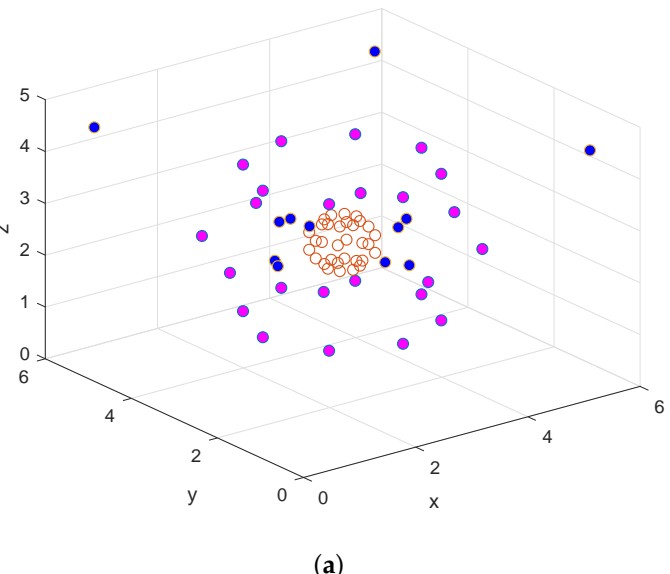

(**a**)

**Figure 5.** *Cont.*

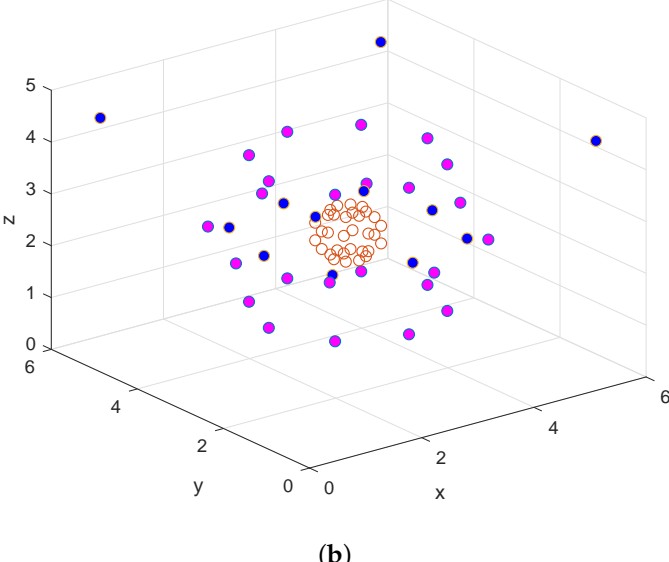

(**b**)

**Figure 5.** Two different array setups, when the noise source moves around a sphere, where in both setups, pink points are the primary source positions, blue points are loudspeaker positions, and red points are microphone positions: (**a**) Case 3; (**b**) Case 4.

Figure 6 demonstrates the noise reduction performance for different source positions. For most of the positions, the WDLS method can achieve slightly better noise reduction than the subspace method. This is because the subspace method only uses the principal components while the WDLS method exploits all the information of the secondary path. Since 8 out of 12 loudspeakers are located in the x-y plane, the noise source positions indicated better noise reduction levels in Figure 6a,b are position No. 2, 5, 8, 11, 14, 17, 20, 23, which are the source candidates on the x-y plane. As the accuracy of the microphone recordings is reduced, the performance prediction becomes less accurate. For example, as shown in Figure 6b, at the noise source position 15, using both methods, the noise reduction levels are positive, which indicates the opposite result from that of Figure 6a.

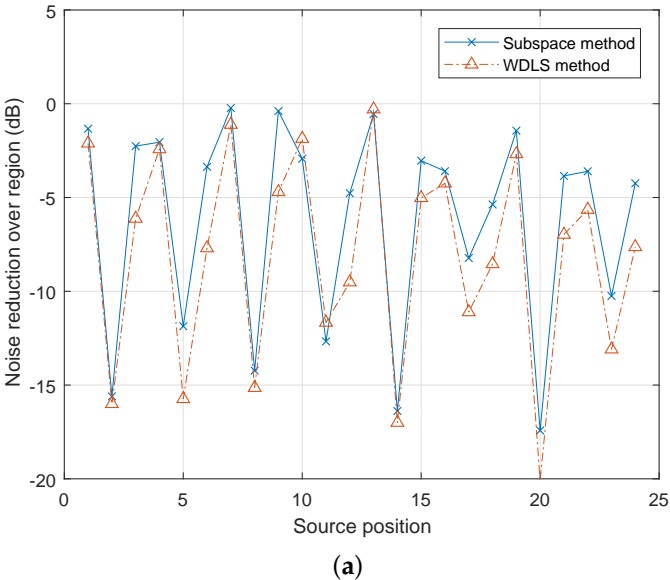

(**a**)

**Figure 6.** *Cont.*

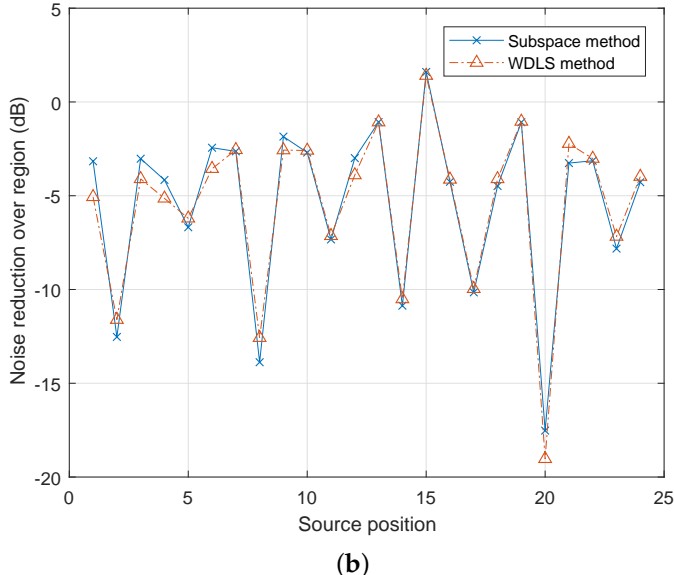

(**b**)

**Figure 6.** Noise reduction performance in case 3, when the noise field is generated by one primary source moving around the sphere using different methods: (**a**) with SNR = 60 dB white noise on the microphone recordings; (**b**) with SNR = 30 dB white noise on the microphone recordings.

Figure 7 demonstrates the energy of the loudspeaker driving signals ($E_d$ in (35)) using different methods (Here, we evaluate the summation of squared driving signals, for all the loudspeakers.). As shown in Figure 7a,b, in both cases, compared with the WDLS method, the proposed subspace method can reduce the total energy on the loudspeakers significantly, which can avoid the overloading of the secondary sources.

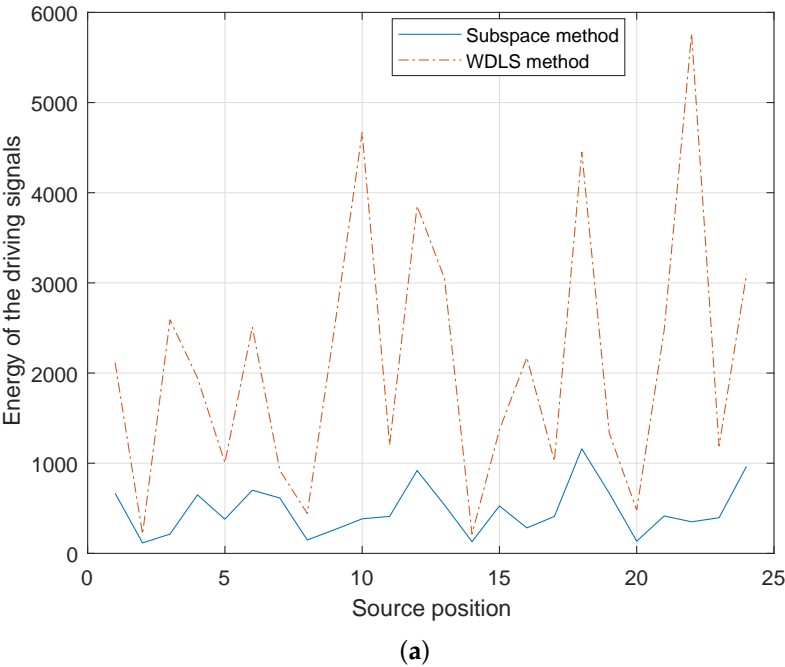

(**a**)

**Figure 7.** *Cont.*

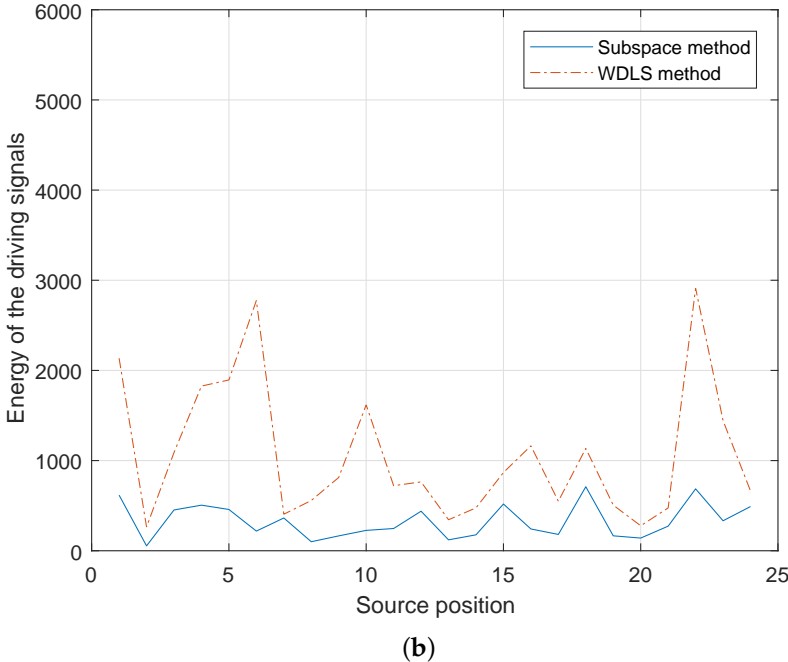

**(b)**

**Figure 7.** Energy of the driving signals in case 3, when the noise field is generated by one primary source moving around the sphere using different methods: (**a**) with SNR = 60 dB white noise on the microphone recordings; (**b**) with SNR = 30 dB white noise on the microphone recordings.

### 5.4. Comparison of the Effect of Different Loudspeaker Placements

Since the loudspeaker placements effect the numbers of principal components in the subspace method, we investigate the noise reduction performance and energy of driving signals under different loudspeaker configurations. The ANC systems with two configurations are shown in Figure 5. In case 4, loudspeakers in the x-y plane have symmetric geometry with respect to the control region, so that the spatial correlation between loudspeakers is greater than that in case 3 (non-symmetry).

Figures 8 and 9 demonstrate the noise reduction and the energy of the loudspeaker driving signals for each noise source positions. For both loudspeaker configurations, compared with the WDLS method, the proposed subspace method achieves less noise reduction and less total energy on the loudspeakers. The significantly reduced energy of the driving signals can avoid the overloading of the secondary sources and avoid the sound amplification outside the control area. In case 3, the correlation between different loudspeakers is higher than that in case 4. In case 4, the number of principal components is larger than that in case 3. Therefore, compared with Figures 8a and 9a, in Figures 8b and 9b, there are less difference between the subspace method and the WDLS method.

### 5.5. Summary and Discussion

In summary, when the number of loudspeakers is less than the number of modes required for representing the primary noise field within the region of interest, ANC over the entire control region cannot be fully achieved. Both WDLS method and subspace method can predict the maximum achievable noise control performance over a spatial region for a given system. In most cases, the WDLS method has slightly better noise reduction performance compared to the subspace method. However, using the subspace method, both the total loudspeaker energy and the energy of the residual noise field outside the control region can be maintained at a relatively low level, which can avoid overloading of the secondary sources and sound amplification outside the control region. From the simulation results, we also notice that if the spatial correlation between different loudspeakers are small, or the accuracy

of the microphone recordings are low, there are less difference between the subspace method and the WDLS method, in terms of noise reduction over the control region and energy of the loudspeakers.

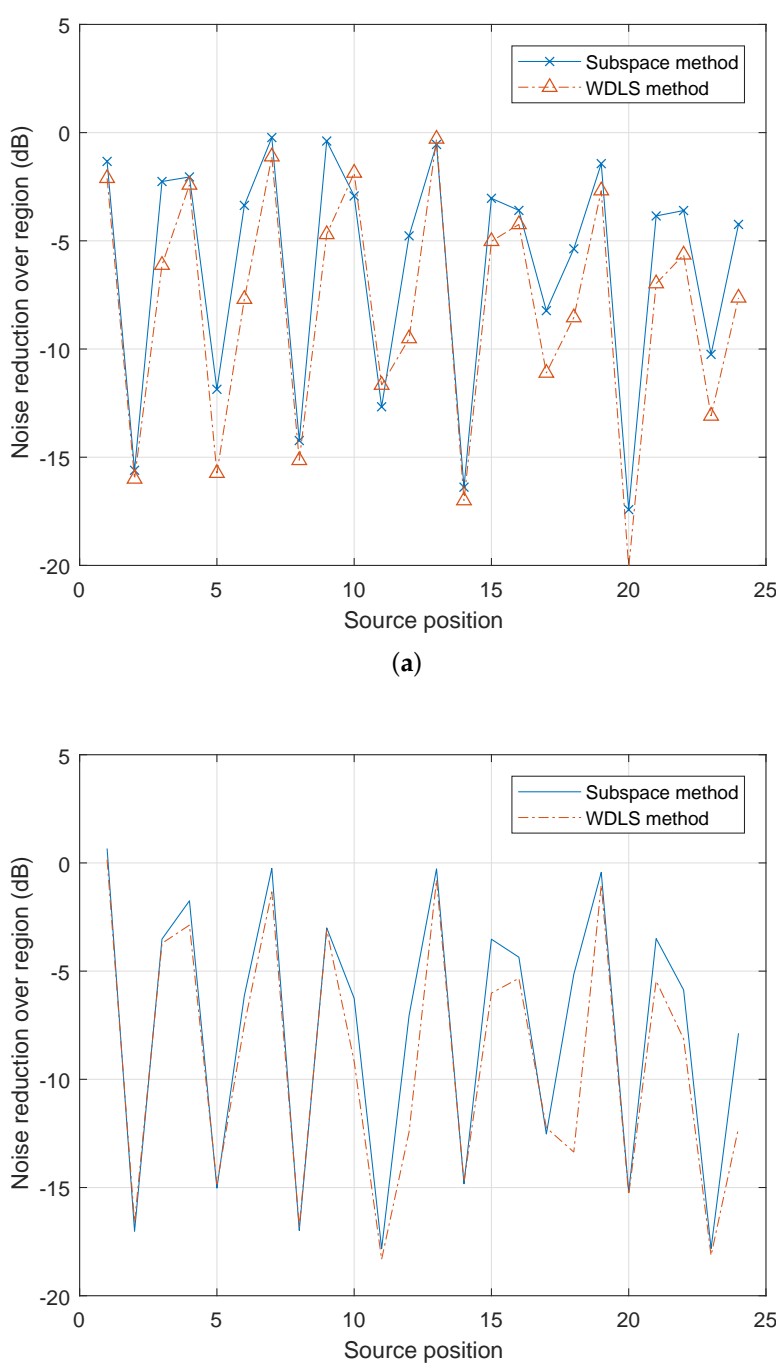

**Figure 8.** Noise reduction over the region using different loudspeaker setups, when the noise field generated by one primary source moving around the sphere using different methods: (**a**) case 3; (**b**) case 4.

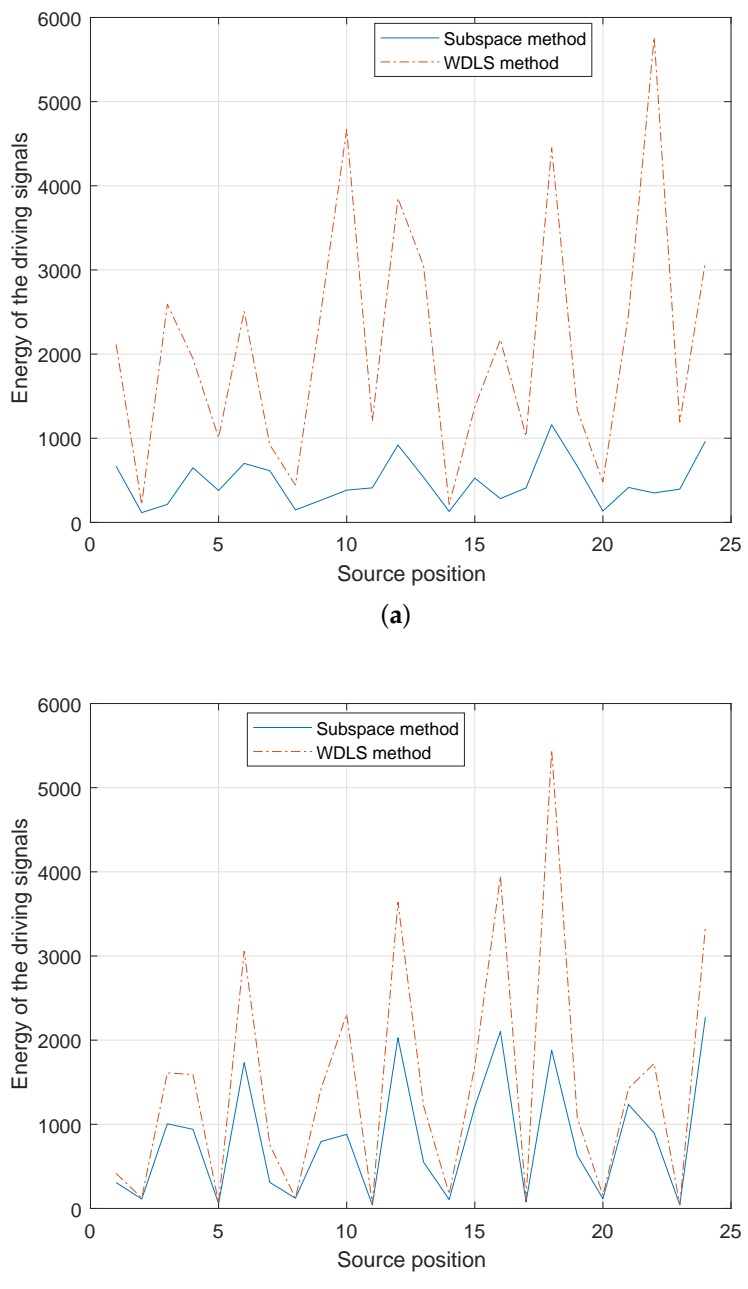

**Figure 9.** Energy of the driving signals generated by one primary source moving around the sphere using different methods: (**a**) case 3; (**b**) case 4.

## 6. Conclusions

In this paper, we analyzed the performance of ANC systems in 3-D reverberant environments, especially when the secondary sources have constraints on numbers and locations. We investigated the maximum achievable ANC performance based on the primary noise field and the secondary-path information in the wave domain.

We discussed a WDLS method to analyze the maximum ANC performance by matching the secondary sound field to the primary noise field in the wave domain. We proposed a subspace method to analyze the maximum ANC performance by investigating the subspace of secondary-path coefficients. We compared the proposed subspace method with the WDLS method under different

loudspeaker configurations and different noise source positions, when the number of secondary sources could not control all the modes in the control region. We validated the noise reduction performance inside the control region, energy of the loudspeakers, and energy of the residual signals outside the control region.

Using the subspace method, we obtained a feasible solution with slightly lower noise reduction level inside the control region, significantly less energy on the loudspeakers, and significantly less energy on the residual noise field outside the control region. The validation with broadband noise signals and the validation with measurements in real spatial ANC applications will be conducted in the future work.

**Author Contributions:** Conceptualization, J.Z., T.D.A., W.Z. and P.N.S.; Funding acquisition, T.D.A., W.Z. and P.N.S.; Investigation, J.Z.; Methodology, J.Z., T.D.A., W.Z. and P.N.S.; Project administration, T.D.A.; Supervision, T.D.A., W.Z. and P.N.S.; Validation, J.Z.; Writing—original draft, J.Z.; Writing—review and editing, T.D.A., W.Z. and P.N.S.

**Funding:** This research was funded by ARC Discovery Project Grant number DP180102375.

**Conflicts of Interest:** The authors declare no conflict of interest.

## Abbreviations

The following abbreviations are used in this manuscript:

| | |
|---|---|
| 3-D | Three-dimensional |
| ANC | Active noise control |
| ATF | Acoustic transfer function |
| PCA | Principal component analysis |
| WFS | Wave field synthesis |
| WD | Wave domain |
| WDLS | Wave-domain least squares method |
| SNR | Signal to noise ratio |

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
