# Peer review of "Active Noise Control over Space: A Subspace Method for Performance Analysis"

_applsci, doi:10.3390/app9061250_

Round 1
Reviewer 1 Report
The presented paper is a very good study about active noise control. The paper is correctly written and all the analysis and math are correctly performed. I just have few minor remarks in order to definitely fit the paper for its publication in the chosen journal:
Avoid the use of “we” in scientific paper.
Why chapter discussion is totally missing? Normally a chapter discussion is included to compare proper results with others in literature.
While dealing with Loudspeaker, please mention “Bianco, F., Teti, L., Licitra, G., & Cerchiai, M. (2017). Loudspeaker FEM modelling: Characterisation of critical aspects in acoustic impedance measure through electrical impedance. Applied Acoustics, 124, 20-29.”
Conclusions are a bit short. Please increase them by shortly report what and how the paper analyses, before really reports what were the results of the paper.
Author Response
The detailed response has been attached.

Reviewer 2 Report
The subject is interesting and original and the paper well describes the proposed subspace method and its effectiveness. The natural completion of the research would be a verification with measurements on a real application of a spatial ANC; do authors think they can do it and describe in a future work?
Only one clarification about the presentation of the results in figures 7 and 9: is the Energy of the driving signals adimensional?
Author Response

(The authors gave the same response as above.)
